# Occupational prestige and sickness absence inequality in employed women and men in Sweden: a registry-based study

Chioma Adanma Nwaru [ID],[1] Tomas Berglund,[2] Gunnel Hensing[1]

[1]School of Public Health and Community Medicine, Institute of Medicine, Sahlgrenska Academy, University of Gothenburg, Goteborg, Sweden
[2]Department of Sociology and Work Science, University of Gothenburg, Goteborg, Sweden

**Correspondence to**
Dr Chioma Adanma Nwaru;
chioma.nwaru@gu.se

## ABSTRACT

**Objectives** Socioeconomic position has been linked to sickness absence (SA). However, less is known about the role of occupational prestige, a measure of social status afforded by one's occupation, in SA. We investigated the association between occupational prestige and SA and the distribution of the association in women and men. We also examined the effect of intersections of gender and occupational prestige on SA.

**Design** Longitudinal.

**Setting** A nationwide representative sample of Swedish working population.

**Participants** 97 397 employed individuals aged 25–59 years selected from the 2004, 2007 and 2010 waves of the Swedish Labour Force Survey and prospectively linked to the Swedish Longitudinal Integration Database for Health Insurance and Labour Market Studies.

**Outcome measures** The number of SA days in any particular year during a 3-year follow-up and long-term (>120 days) SA based on those with at least one sick leave spell during the follow-up.

**Results** Occupational prestige was weakly associated with SA in the total sample after adjusting for potential confounders. In the gender-stratified analysis, women in lower prestige occupations had higher absenteeism rates than women in high prestige occupations; men in lower prestige occupations had higher odds for long-term SA than men in high prestige occupations. In the intersectional analysis, women regardless of prestige level and men in lower prestige occupations had higher probability of SA compared with men in high prestige occupations. Women in high prestige occupations had the highest absenteeism rates (incidence rate ratio (IRR), 2.25, 95% CI, 2.20 to 2.31), while men in medium prestige occupations had the lowest rates (IRR, 1.17, 95% CI, 1.13 to 1.20). Compared with the rest of the groups, men in low and medium prestige occupations had higher odds for long-term absence.

**Conclusion** There is need to pay close attention to occupational prestige as a factor that may influence health and labour market participation.

## INTRODUCTION

Sickness absence (SA), particularly long-term absence, has consistently been associated with unemployment,[1] disability retirement[2] and

### Strengths and limitations of this study

► A large, reliable, nationally representative sample of employed women and men in Sweden was used for the study.
► Data on sickness absence (SA) was register-based.
► The incorporation of gender-occupational prestige intersections expanded previous knowledge on gender differences in SA.
► Occupational prestige was based on job titles provided in the inclusion year, which may not necessarily be the longest held occupations for some of the participants.
► The study did not include SA lasting 14 days or less.

mortality,[3] and could lead to permanent exit from the workforce. Socioeconomic position (SEP) is an important determinant of SA. Occupation, income and education are key indicators of SEP. Compared with education and income, occupation is a more complex factor; it does not have a single measure that captures the occupation–health relationship.[4] Previous studies on occupation and SA have used occupational classifications based on employment relations[5] and job titles.[6] Other studies have combined job titles, education and similarity of work content.[7] Several occupational classification schemes exist that have not been studied in relation to SA. Occupation-based measures originate from different theoretical concepts, and each classification tends to capture a specific aspect of occupation related to health. Moreover, the mechanisms connecting occupation to health are not similar across different occupation-based measurements.[4 8] To better understand the role of various aspects of occupation on SA, which is important for creating interventions to reduce SA, an exploration of different occupation-based measures, particularly those that have received less attention, is warranted.

The present study focuses on the role of occupational prestige as a determinant of SA. Occupational prestige as used here reflects social status, given that the measure represents the value placed on different occupations by the society. Occupational prestige is measured using the Standard International Occupational Prestige Scale (SIOPS), which is a scale that provides hierarchical ranking from least to most esteemed occupations according to average societal ratings.[4 9] No studies so far have investigated the association between occupation and SA using SIOPS-based occupational prestige as an indicator. Investigating this association can help to capture features of social hierarchy (acceptance, respect, recognition, admiration, autonomy, power, social network and social support)[10 11] that may not be adequately represented in other measures of occupational classification, such as occupational class, thus illuminating how social position and its associated rewards and privileges contribute in shaping health risk.

The position a given occupation has on the prestige scale strongly impacts both the adoption of an occupational identity and construction of self-esteem.[12–14] Compared with individuals in lower prestige occupations, those in high prestige occupations are more likely to identify with their occupations because of the social rewards and privileges associated with high status occupations.[12 13] Owing to a lack of such rewards and privileges, employees in lower prestige occupations may have increased risk of work alienation (ie, psychological detachment from their jobs), low self-esteem, poor work satisfaction and reduced social interactions.[13 14] These negative affects can induce poor response to stress and consequently increase the risk of poor health.[15 16] Previous research has shown that occupational prestige is a determinant of different health outcomes, and that it can influence health independently of other SEP indicators and job characteristics. For example, Fujishiro et al[17] reported increased poor self-rated health among employees in lower prestige occupations after controlling for occupational categories, education, income, job stress, workplace social support and job satisfaction. Additionally, occupational prestige was found to be a determinant of self-rated health[18] as well as a determinant of mortality,[19] even when controlling for education and income in both studies. In line with these studies, we expect to find increased risk of SA among employees working in lower prestige occupations.

Gender is an important determinant of SA. A common statistical approach when investigating gender differences in SA is to compare the risk in women and men, while controlling for potential confounders. Thus, in several studies, women (and men) are treated as a homogeneous group.[20 21] Hankivsky et al[22] noted that such an approach fails to recognise the diversity of women (and men) regarding other forms of social identities and the relational nature among different social identities. They suggested that researchers incorporate intersectional approach, that is, investigate the extent to which gender combines with other social identities to create health inequality. The intersectional approach integrates both between and within group differences,[23] therefore offering the potential to provide insight on different levels of SA inequality.

Our aims in this study are to (1) investigate whether occupational prestige is associated with SA, (2) examine the distribution of this association among women and men, and (3) investigate the effects of intersections of gender and occupational prestige on SA. The current study has the potential to improve knowledge on determinants of SA as well as on target groups, both of which are crucial for reducing SA and improving labour market participation.

## METHODS

### Study population and design

This study is part of the New Ways and the Polarization programmes at the University of Gothenburg. The study is based on data of participants in the Swedish Labour Force Survey (LFS), which is a quarterly survey used to monitor labour market developments for the entire population aged 15–74 years. The LFS uses a rotating sampling design, meaning that participants in the survey are interviewed once every quarter for a maximum of eight times over a 2-year period, after which they are replaced with new sample individuals.[24] For this study, we selected from three cross-sections at the end of the survey years 2004, 2007 and 2010, individuals aged 25–59 years who were employed wage earners according to their main economic activity and were at work, that is, not self-employed, unemployed, retired or on disability pension (N=107 608). Using the participants' encrypted identification number, we linked information from the cohorts with the Swedish Longitudinal Integration Database for Health Insurance and Labour Market Studies (LISA) register[25] and followed each cohort for 3 years thus covering the period 2005–2013. We excluded from the cohorts those participants who started receiving old age or disability pension by the end of the study period (n=6278). We also excluded participants with incomplete follow-up data and those with missing exposure or potential control variables (n=3933), resulting in a final working sample of 97 397 participants (figure 1).

### Patient and public involvement

This study was performed without patient or public involvement. The Swedish national registers are protected by special legislation that makes it possible for researchers to collect certain information without personal consent.

### Measurements

#### Occupational prestige

We measured occupational prestige, the exposure in this study, with the SIOPS.[4 9] The scores, which ranged from 13 to 78, were calculated based on occupational prestige studies conducted in more than 60 countries. Respondents in each of the participating countries were asked to rate a set of occupational titles with respect to their

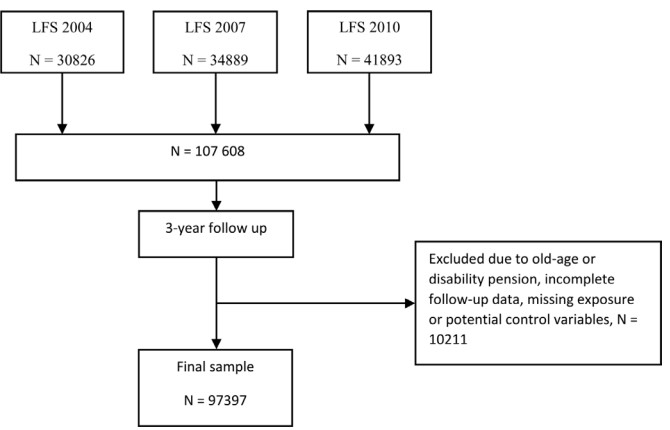

**Figure 1** Flowchart illustrating selection of study participants. LFS, labour force survey.

social position. The ratings were aggregated into mean scores and reported as indicators of the relative prestige scores of the evaluated occupations.[26] Svensson and Ulfsdotter[27] compared the SIOPS and occupational prestige scores generated from a Swedish population and found a high correlation (0.8) between the scores. For this study, we produced the prestige scores by assigning an occupational prestige score (ie, the SIOPS) to each occupational title provided by the respondents (obtained from the LISA) during the inclusion (ie, 2004, 2007 and 2010) based on a four-level International Standard Classification of Occupations, revision 1988, classification. Based on the distribution of occupational prestige in the current data, we divided the scores into tertiles (15–34 points, 35–50 points and 51–78 points) to represent low, medium and high occupational prestige categories, respectively. The

last category was used as the reference in the analyses. We constructed six intersectional groups using information on the prestige groups and on gender (obtained from the LFS). The groups were women in low prestige occupations, women in medium prestige occupations, women in high prestige occupations, men in low prestige occupations, men in medium prestige occupations and men in high prestige occupations (reference group). Table 1 provides the five most common high prestige scores and examples of occupations in each prestige category for women and men.

### Sickness absence

In Sweden, all individuals aged 16 years and above with an income from work or unemployment benefits are entitled to SA benefits. If the sick leave lasts for more than 7 consecutive days, starting from the eighth day, a doctor's certificate is required. During the study observation years (2005–2013), the employers paid for absence days for 14 days or less after a qualifying day without benefits. From day 15 onward, the Social Insurance Agency (SIA) pays SA benefits. Unemployed individuals have one qualifying day and receive sick pay from the SIA from the second day. In this study, information on SA was based on data from the LISA that only contains information on SA from day 15 onward. We used two measures of SA: the number of SA days in any particular year during the follow-up and long-term SA. The latter was based on those with at least one sick leave spell during the follow-up and defined as absence days lasting over 120 days. In the so-called rehabilitation chain in Sweden, evaluation of work capacity and the right to SA benefit are assessed at specific times, starting from day 90 into the SA period. By choosing 120

| | Examples of common occupations in each category according to International Standard Classification of Occupations, revision 1988 |
|---|---|
| **Table 1** List of five most common high prestige scores and examples of occupations in each prestige category for women and men | |
| Five most common high prestige scores (SIOPS) among women | |
| 52 | Social work professions, life science technicians and managers of small enterprises in health and social work |
| 53 | Administrative secretaries and related associate professionals, and computer assistants |
| 54 | District nurses, midwives, head nurses, Liberians and related information professionals |
| 57 | Primary education teaching professionals, market research analysts and related professionals |
| 60 | Teaching professionals, artistic and practical subjects, teaching professionals, academic subjects, production and operations managers in education |
| Five most common high prestige scores (SIOPS) among men | |
| 51 | Computer systems designers, analysts and programmers, and computing professionals not elsewhere classified |
| 53 | Computer assistants, administrative secretaries and related associate professionals |
| 57 | Primary education teaching professionals, business professionals not elsewhere classified, market research analysts and related professionals |
| 60 | Teaching professionals, artistic and practical subjects, teaching professionals, academic subjects, finance and administrative managers, sales and marketing managers, and vocational teaching professionals |
| 78 | Medical doctors, college, university and higher education teaching professionals |

SIOPS, Standard International Occupational Prestige Scale.

days as the cut point when the individuals would have undertaken the first assessment, our measure of long-term SA is more likely to represent serious health conditions, which are of interest in this study.

## Potential confounders

From the LFS database, we extracted information on participants' age (25–34, 35–44 and 45–59 years), marital status (single, cohabiting and married), occupational class (manual, lower non-manual and higher non-manual), employment sector (private or public), employment type (temporary or permanent), and contract type (full-time or part-time) at the time of inclusion in the study. From the LISA register, we obtained information on previous SA for the 2 years preceding the study observation period. We also retrieved information on highest education attained (primary, secondary or tertiary) and on gross annual salaries (divided into quartiles) of the participants during the inclusion year from the LISA database. We categorised the year of measurement as 2004, 2007 and 2010. We treated these variables as confounders capable of influencing the association between occupational prestige and SA.

## Statistical analyses

Data analyses were performed with the generalised estimating equation (GEE), a statistical technique suitable for modelling repeated outcome variables. The technique takes into account the within-subject correlation of the repeated SA variable.[28] The annual rate of SA days was estimated using GEE with negative binomial. The negative binomial distribution was chosen because of the overdispersion in the count outcome variable.[29] Exchangeable correlation was specified in all the models as the correlation structure based on the result of a preliminary investigation. However, GEE is robust to misspecification of the correlation structure.[28] Six separate models were estimated, starting with an age-adjusted model followed by five other adjusted models. In Model I, survey year, gender and marital status were entered into the model. In Model II, previous SA was additionally included into the model. In order to estimate the effect of occupational prestige on SA in the presence of other commonly used SEP indicators, additional adjustment was made for income and education in Model III, and occupational class in Model IV. Employment-related characteristics, which we interpreted as proxy measures for work-related characteristics, were added in the final model (Model V). The coefficients were expressed as incidence rate ratios (IRRs) and 95% CIs.

GEE with logistic regression was used to estimate the annual risk of having >120 SA days during the 3-year follow-up. The modelling strategy was the same as for the count outcome. The robustness of the results regarding long-term SA was tested by repeating the analyses using 90 days as the cut point for long-term SA since the Swedish evaluation of work capacity within the rehabilitation framework starts at day 90. All the analyses were

population weighted to correct for biases due to sampling. IBM SPSS V.26 for Windows was used for all the statistical analyses.

## RESULTS

### Descriptive statistics

The mean age of the study population was 42 years (SD, 9.7 years). The participants' characteristics, overall and by occupational prestige groups, are presented in table 2. There were more women (than men) in low prestige occupations; the same applied for temporary (vs permanent) and part-time (vs full-time) employees, and those with previous SA (vs those without). Working in high prestige occupations was prevalent among those in higher non-manual occupations, those with a tertiary education, and those in the highest income quartiles. During the study follow-up, 22% (21 836) of the participants had at least one episode of SA lasting more than 14 days. Sixteen per cent (n=3709) of the 21 836 had at least one episode of SA lasting over 120 days. All participants' characteristics were associated with both the number of SA days and long-term absence at follow-up (table 2).

### Occupational prestige differences in number of sickness absence days

Table 3 shows the IRRs for the association between occupational prestige and number of SA days in the total sample. There was a graded association between occupational prestige and SA in the age-adjusted model. Employees in low prestige occupations had over two times more SA than those in high prestige group. Those in medium prestige occupations had 1.41 times more SA than employees in high prestige group. The IRRs increased after adding survey year, gender and marital status into the model. Additional adjustment for previous SA attenuated the IRRs for both groups. Additional adjustment for income and education increased the IRR for employees in low prestige occupations and significantly attenuated the IRR for employees in medium prestige occupations. Additional adjustment for occupational class eliminated the increased IRR among employees in lower prestige occupations. Adjustment for employment-related variables did not seem to have much influence on the association after adjustment for sociodemographic and socioeconomic factors.

When the data were analysed separately for women and men, the result from the age-adjusted model (online supplemental table A1) showed significant occupational prestige differences in both genders. As with the total sample, additional adjustment for income and education (Model III) and occupational class (Model IV) significantly reduced the estimates. After adjusting for all potential confounders (Model V), among women, the IRR remained statistically significant among women in low (IRR, 1.22, 95% CI, 1.18 to 1.27) and medium (IRR, 1.05, 95% CI, 1.02 to 1.07) occupational prestige groups compared with women in the high occupational prestige

**Table 2** Distribution of sample characteristics by occupational prestige at baseline and the association of the characteristics with SA at follow-up

| | Total N=97 397 | Occupational prestige at baseline | | | | IRR associated with number of SA days at follow-up N=97 397 | OR associated with long-term SA at follow-up N=21 836* |
|---|---|---|---|---|---|---|---|
| | n (weighted %) | Low (n=32 343) weighted % | Medium (n=34 556) weighted % | High (n=30 498) weighted % | P value | IRR (95% CI) | OR (95% CI) |
| Survey year | | | | | 0.000 | | |
| 2004 | 27 950 (34.1) | 32.2 | 36.7 | 31.0 | | 1.32 (1.31 to 1.34) | 1.48 (1.45 to 1.50) |
| 2007 | 31 483 (32.8) | 32.8 | 35.3 | 31.9 | | 0.88 (0.87 to 0.89) | 0.97 (0.96 to 0.99) |
| 2010 | 37 964 (33.1) | 32.5 | 34.9 | 32.6 | | 1.00 | 1.00 |
| Age (years) | | | | | 0.000 | | |
| 25–34 | 26 259 (27.7) | 34.2 | 35.3 | 30.5 | | 1.00 | 1.00 |
| 35–44 | 30 890 (32.1) | 31.0 | 36.2 | 32.8 | | 1.32 (1.30 to 1.34) | 1.38 (1.35 to 1.41) |
| 45–59 | 40 248 (40.2) | 32.5 | 35.5 | 32.0 | | 1.63 (1.61 to 1.65) | 1.49 (1.46 to 1.51) |
| Gender | | | | | 0.000 | | |
| Women | 48 029 (47.5) | 34.8 | 32.1 | 33.1 | | 1.87 (1.85 to 1.89) | 1.11 (1.09 to 1.13) |
| Men | 49 368 (52.5) | 30.4 | 38.9 | 30.7 | | 1.00 | 1.00 |
| Marital status | | | | | 0.000 | | |
| Single | 23 441 (24.3) | 37.2 | 34.8 | 28.0 | | 1.25 (1.24 to 1.26) | 1.29 (1.27 to 1.31) |
| Cohabiting | 28 386 (28.9) | 33.7 | 37.6 | 28.7 | | 0.93 (0.92 to 0.94) | 0.95 (0.93 to 0.96) |
| Married | 45 570 (46.8) | 29.3 | 34.9 | 35.8 | | 1.00 | 1.00 |
| Education | | | | | 0.000 | | |
| Primary | 9668 (10.0) | 59.5 | 33.5 | 7.0 | | 2.16 (2.13 to 2.20) | 1.56 (1.53 to 1.59) |
| Secondary | 46 415 (47.2) | 47.2 | 39.7 | 13.1 | | 1.57 (1.55 to 1.59) | 1.21 (1.19 to 1.23) |
| Tertiary | 41 314 (42.7) | 10.0 | 31.7 | 58.4 | | 1.00 | 1.00 |
| Occupational class | | | | | 0.000 | | |
| Manual | 40 389 (40.8) | 73.7 | 26.2 | 0.2 | | 1.00 | 1.00 |
| Lower non-manual | 38 522 (39.5) | 6.0 | 61.2 | 32.8 | | 0.58 (0.57 to 0.58) | 0.75 (0.73 to 0.76) |
| Higher non-manual | 18 486 (19.7) | 0.1 | 4.2 | 95.7 | | 0.38 (0.37 to 0.38) | 0.70 (0.69 to 0.72) |
| Individual income | | | | | 0.000 | | |
| Lowest | 24 365 (25.2) | 54.3 | 29.5 | 16.2 | | 5.63 (5.54 to 5.71) | 3.40 (3.31 to 3.49) |
| Second quartile | 24 369 (24.8) | 41.0 | 38.7 | 20.3 | | 3.12 (3.07 to 3.17) | 1.92 (1.87 to 1.97) |
| Third quartile | 24 330 (24.5) | 26.1 | 42.1 | 31.9 | | 2.18 (2.15 to 2.22) | 1.58 (1.53 to 1.62) |
| Highest | 24 333 (25.5) | 8.9 | 32.6 | 58.5 | | 1.00 | 1.00 |
| Employment sector | | | | | 0.000 | | |
| Private | 63 136 (66.5) | 31.9 | 41.5 | 26.6 | | 1.00 | 1.00 |
| Public | 34 261 (33.5) | 33.7 | 24.0 | 42.3 | | 1.65 (1.63 to 1.67) | 1.27 (1.25 to 1.29) |
| Employment type | | | | | 0.000 | | |
| Temporary | 8764 (9.1) | 41.5 | 25.9 | 32.6 | | 1.14 (1.12 to 1.16) | 1.17 (1.15 to 1.20) |
| Permanent | 88 633 (90.9) | 31.6 | 36.6 | 31.8 | | 1.00 | 1.00 |
| Contract type | | | | | 0.000 | | |
| Part-time | 19 641 (19.3) | 49.6 | 27.4 | 23.0 | | 1.60 (1.58 to 1.61) | 1.33 (1.31 to 1.35) |
| Full-time | 77 756 (80.7) | 28.4 | 37.6 | 34.0 | | 1.00 | 1.00 |
| Previous SA in the last two years | | | | | 0.000 | | |
| No | 81 995 (84.8) | 30.4 | 36.1 | 33.6 | | 1.00 | 1.00 |
| Yes | 15 402 (15.2) | 44.4 | 33.4 | 22.3 | | 5.04 (4.99 to 5.09) | 2.45 (2.42 to 2.49) |

*Analyses performed among those with at least one spell of registered SA during follow-up.
IRR, incidence rate ratio; SA, sickness absence.

**Table 3** Association of occupational prestige with number of sickness absence days at follow-up. IRRs and 95% CIs obtained from generalised estimating equation with negative binomial regression

| | Total N=97 397 | Age-adjusted | Model I | Model II | Model III | Model IV | Model V |
|---|---|---|---|---|---|---|---|
| | Weighted % | IRR (95% CI) | IRR (95% CI) | IRR (95% CI) | IRR (95% CI) | IRR (95% CI) | IRR (95% CI) |
| Occupational prestige | | | | | | | |
| Low | 32.5 | 2.48 (2.45 to 2.51) | 2.68 (2.64 to 2.71) | 2.35 (2.32 to 2.39) | 1.44 (1.41 to 1.47) | 1.01 (0.98 to 1.04) | 1.02 (1.00 to 1.05) |
| Medium | 35.7 | 1.41 (1.40 to 1.43) | 1.61 (1.59 to 1.64) | 1.53 (1.50 to 1.55) | 1.12 (1.11 to 1.14) | 0.96 (0.94 to 0.98) | 1.00 (0.98 to 1.02) |
| High | 31.8 | 1.00 | 1.00 | 1.00 | 1.00 | 1.00 | 1.00 |

Model I, additionally adjusting for survey year, gender and marital status. Model II, additionally adjusting for previous sickness. Model III, additionally adjusting for education and income. Model IV, additionally adjusting for occupational class. Model V, additionally adjusting for employment type, contract type and employment sector.
IRR, incidence rate ratio.

group. Among men, no increased IRR was found both for men in low (IRR, 1.00, 95% CI, 0.96 to 1.05) and medium (IRR, 0.92, 95% CI, 0.89 to 0.95) prestige occupations compared with men in high prestige occupations.

Table 4 presents the distribution (weighted %) of intersectional groups and the IRR across the groups, with men in the high occupational prestige group as the reference group. All the groups had increased absence rates compared with men in high prestige occupations in the age-adjusted model. After accounting for all potential confounders, women in high prestige occupations had the highest absence rates (IRR, 2.25, 95% CI, 2.20 to 2.31), which is an unexpected finding in this study. Women in medium prestige occupations had the second highest absence rate (IRR, 2.01, 95% CI, 1.95 to 2.06) followed by women in low prestige occupations (IRR, 1.68, 95% CI, 1.62 to 1.74), men in low prestige occupations (IRR, 1.37, 95% CI, 1.32 to 1.42) and men in medium prestige occupations (IRR, 1.17, 95% CI, 1.13 to 1.20).

### Occupational prestige differences in long-term sickness absence

In the total sample, employees in lower prestige occupations had higher OR for long-term SA than those in high prestige occupations in the age-adjusted model (online

supplemental table A2). However, after adjusting for all potential confounders, only the OR for employees in low prestige occupations showed slightly increased OR for long-term SA (OR, 1.07, 95% CI, 1.04 to 1.11). In the gender-stratified results seen in table 5, among men, there was a strong graded association between occupational prestige and long-term SA in the age-adjusted model. Additional adjustment for survey year and marital status (Model I), previous SA (Model II), as well as education and income (Model III) had an attenuating effect, while adjustment for occupational class (Model IV) increased the ORs. After accounting for all potential confounders, men in low and medium prestige occupations had 26% and 14% increased odds, respectively, compared with men in high prestige occupations (Model V). Among women, there were no occupational prestige differences and no increased ORs with respect to long-term SA after accounting for all potential confounders (Model V).

The results with the intersectional groups showed age-adjusted ORs ranging from 1.43 to 2.09, with the highest OR found among women in low prestige occupations (online supplemental table A3). However, in the full model (Model V), men in low prestige occupations had the highest risk (OR, 1.31, 95% CI, 1.25 to 1.38), followed

**Table 4** Intersections of gender and occupational prestige and the association with number of sickness absence days at follow-up. IRRs and 95% CIs obtained from generalised estimating equation with negative binomial regression

| | Total N=97 397 | Age-adjusted | Model I | Model II | Model III | Model IV | Model V |
|---|---|---|---|---|---|---|---|
| | Weighted % | IRR (95% CI) | IRR (95% CI) | IRR (95% CI) | IRR (95% CI) | IRR (95% CI) | IRR (95% CI) |
| Women/low prestige occup. | 16.5 | 6.06 (5.92 to 6.21) | 6.18 (6.04 to 6.32) | 4.91 (4.80 to 5.02) | 2.56 (2.49 to 2.64) | 1.76 (1.70 to 1.82) | 1.68 (1.62 to 1.74) |
| Women/medium prestige occup. | 15.2 | 3.59 (3.50 to 3.68) | 3.58 (3.50 to 3.67) | 3.00 (2.93 to 3.08) | 2.11 (2.06 to 2.17) | 1.98 (1.93 to 2.03) | 2.01 (1.95 to 2.06) |
| Women/high prestige occup. | 15.7 | 2.93 (2.86 to 3.01) | 2.99 (2.92 to 3,06) | 2.68 (2.61 to 2.74) | 2.38 (2.32 to 2.43) | 2.35 (2.30 to 2.41) | 2.25 (2.20 to 2.31) |
| Men/low prestige occup. | 16.0 | 3.60 (3.51 to 3.69) | 3.57 (3.48 to 3.65) | 3.11 (3.04 to 3.19) | 1.93 (1.88 to 1.99) | 1.34 (1.29 to 1.38) | 1.37 (1.32 to 1.42) |
| Men/medium prestige occup. | 20.4 | 2.15 (2.10 to 2.20) | 2.15 (2.10 to 2.21) | 2.06 (2.01 to 2.11) | 1.43 (1.39 to 1.47) | 1.12 (1.09 to 1.15) | 1.17 (1.13 to 1.20) |
| Men/high prestige occup. | 16.1 | 1.00 | 1.00 | 1.00 | 1.00 | 1.00 | 1.00 |

Model I, additionally adjusting for survey year and marital status. Model II, additionally adjusting for previous sickness. Model III, additionally adjusting for education and income.
Model IV, additionally adjusting for occupational class. Model V, additionally adjusting for employment type, contract type and employment sector
IRR, incidence rate ratio.

**Table 5** Gender-stratified association between occupational prestige and long-term SA among those with at least one spell of SA. ORs and 95% CIs obtained from generalised estimating equation with logistic regression

| | | Age-adjusted OR (95% CI) | Model I OR (95% CI) | Model II OR (95% CI) | Model III OR (95% CI) | Model IV OR (95% CI) | Model V OR (95% CI) |
|---|---|---|---|---|---|---|---|
| Women | N=13 719 weighted % | | | | | | |
| Occupational prestige | | | | | | | |
| Low | 43.7 | 1.40 (1.37 to 1.43) | 1.39 (1.36 to 1.42) | 1.30 (1.27 to 1.33) | 1.06 (1.03 to 1.09) | 1.01 (0.96 to 1.05) | 0.98 (0.94 to 1.03) |
| Medium | 29.6 | 1.03 (1.00 to 1.05) | 1.01 (0.99 to 1.04) | 0.99 (0.96 to 1.01) | 0.90 (0.87 to 0.92) | 0.95 (0.93 to 0.98) | 0.98 (0.95 to 1.01) |
| High | 26.7 | 1.00 | 1.00 | 1.00 | 1.00 | 1.00 | 1.00 |
| Men | N=8117 weighted % | | | | | | |
| Occupational prestige | | | | | | | |
| Low | 43.0 | 1.92 (1.85 to 1.99) | 1.86 (1.79 to 1.93) | 1.64 (1.58 to 1.71) | 1.07 (1.02 to 1.12) | 1.26 (1.19 to 1.34) | 1.26 (1.19 to 1.34) |
| Medium | 39.4 | 1.42 (1.37 to 1.48) | 1.40 (1.35 to 1.46) | 1.28 (1.23 to 1.33) | 0.97 (0.92 to 1.01) | 1.12 (1.06 to 1.19) | 1.14 (1.08 to 1.21) |
| High | 17.6 | 1.00 | 1.00 | 1.00 | 1.00 | 1.00 | 1.00 |

Model I, additionally adjusting for survey year and marital status. Model II, additionally adjusting for previous sickness. Model III, additionally adjusting for education and income. Model IV, additionally adjusting for occupational class. Model V, additionally adjusting for employment type, contract type and employment sector.
SA, sickness absence.

by men in medium prestige occupations (OR, 1.19, 95% CI, 1.13 to 1.24), women in high prestige occupations (OR, 1.06, 95% CI, 1.02 to 1.11), and women in low prestige occupations (OR, 1.05, 95% CI, 1.00 to 1.11). No statistically significant OR was found among women in medium prestige occupations (OR, 1.02, 95% CI, 0.98 to 1.06). The estimates pointed toward the same direction when we repeated the analyses using 90 days as the cut point (data not shown).

## DISCUSSION

We found a weak association between occupational prestige and SA in the total sample after controlling for age, survey year, gender, marital status, previous SA, education, income, occupational class and employment-related variables. However, when we assessed the distribution of the association separately in women and men, we found a graded association between occupational prestige and SA in both genders. Women in lower prestige occupations had higher rates of SA days than women in high prestige occupations. Men in lower prestige occupations had higher odds for long-term SA than men in high prestige occupations. When we pooled the sample and investigated the effect of intersections of gender and occupational prestige on SA, we found that men in high prestige occupations had the lowest IRR for SA days, as well as the lowest OR for long-term SA than the rest of the groups. Of all the groups, women in high prestige occupations had the highest IRR for SA days, while men in low prestige occupations had the highest odds for long-term SA.

Our finding of increased SA among women and men in lower occupational prestige corroborates previous studies that used different health outcomes.[17–19] Occupational class, income and education significantly impacted on the effect estimates, but the association remained statistically significant, thus supporting the notion that occupational prestige taps into aspects of the occupation–health relationship that are not represented by education, income

and occupational class.[17] Work stress has been hypothesised as one mechanism connecting lower occupational prestige and poor health outcomes. Hoven et al[30] reported that employees in lower prestige occupations had increased work stress, which consequently increased their risk of depressive symptoms. Working in lower prestige occupations may also induce feelings of low self-esteem[13 17] that may lead to increased risk of anxiety, poor social functioning, risky behaviour[31] and poor recovery from illness.[32] Matthews et al[33] found that employees in lower prestige occupations more often experienced interpersonal conflicts, boredom and increased heart rate; and interpersonal conflicts at work have been associated with increased risk of prolonged fatigue and poor general health.[34] This provides another possible reason for the increased SA among employees in lower prestige occupations.

We found increased IRR for SA days across all the intersectional groups compared with men in high prestige occupations, which was expected. Contrary to expectations, the IRR for SA days was higher for women in high prestige occupations than for women and men in lower prestige occupations. This finding supports the notion that some people occupy intersectional locations that include both privileges and marginalisation.[35] The finding highlights the importance of the intersectional approach and the limitations associated with taking a unitary approach in evaluating health inequality.[36] The increased absence rate among women in high prestige occupations could reflect stressful conditions emanating from the combination of household chores and high workload. Berntsson et al[37] found that women in high positions in Sweden are still mainly responsible for household tasks and childcare duties, and they reported higher levels of stressful conditions than men in similar positions.

In this study, among men, we found substantial occupational prestige differences in long-term SA, whereas among women, there was no such differences. In the intersectional

analysis, men in lower prestige occupations also had higher odds for long-term SA than women regardless of prestige levels. Higher SA incidence in women than in men has been well supported in previous studies,[20][21] but longer absence spells among women has been a controversial topic in the literature, including in studies with similar cut-off points. For instance, Lidwall and Marklund[38] reported a higher risk for more than 60 absence days among women compared with men, whereas Laaksonen *et al*[39] did not find increased absence among women using a cut-off point of 60 days. In Sweden[40] and abroad,[41] longer absence duration has been reported among men compared with women, which aligns with our finding of increased risk among men in lower prestige occupations. A possible explanation for this finding may relate to men's value system. Men attribute significant importance to values, such as power, status and achievement.[42] Because they are sensitive to being relegated to a subordinate position, men in lower status occupations may develop negative perceptions of themselves, and to compensate, may engage in unhealthy behaviours that may jeopardise their psychological health.[43] Peterson[44] found that how men perceived their level of control, authority and status had a significant impact on their psychological well-being, and psychiatric illness has been reported previously as a determinant of long-term SA among men in Sweden.[40] Research has also shown that men are more likely to dismiss their health needs and delay seeking healthcare compared with women;[45] such behaviours may worsen health situations and lead to long-term SA. The tendency of women to be more proactive in consulting professional help, may, on the one hand, explain their increased absenteeism rates, and, on the other hand, provide an alternative explanation for their decreased risk for long-term SA as observed in the current study.

The strengths of the current study include the longitudinal design, the use of register-based SA data and the relatively long follow-up (observation period covering 2005–2013), which spanned periods of varying economic situations. The use of GEE made it possible to account for within-individual correlation in the SA variable, which is not attainable using ordinary regression methods. The application of the intersectional approach can be considered as an extension of previous studies on gender differences in SA, because it provided deeper insights on how gender and occupational prestige are mutually constituted to shape SA inequality in the working population.[23] This study was based on data of a large, reliable nationally representative sample (>70% LFS response rate) of employed women and men, making it possible for the findings to be generalised to the Swedish working population. One limitation of the study relates to our measure of occupational prestige. The measure was based on occupational titles provided in the inclusion year, which may not necessarily be the longest held occupations for some of the participants. Due to reliance on occupational data provided in the inclusion year, we could not assess whether a participant changed occupations during follow-up and whether such changes might have any influence on the associations. Previous research has shown increased levels of physical workload among employees in lower social positions (measured as a combination of educational level and occupational groups) that consequently was associated with increased risk of SA.[46] Due to a lack of data, we could not assess the influence of physical workload on the association between occupational prestige and SA. We could not also assess the influence of work-related psychosocial factors (work demand, control and support), which have been associated with occupational prestige[30] and SA.[47] However, we did include several employment-related factors (contract type, employment sector and employment type) as proxy measures for these psychosocial work characteristics, and our findings seem not to be influenced by these factors. Our measure of SA did not include absence spells lasting 14 days or less. We also did not examine cause-specific SA. It is possible that occupational prestige will exert varying effects on these forms of SA.

## CONCLUSION

This research demonstrated an association between lower occupational prestige and increased SA in women and men. Compared with men in high prestige occupations, women in all prestige levels and men in the lower prestige occupations had higher probability of SA days, with the highest IRR found among women in high prestige occupations. Men in low and medium prestige occupations had higher odds for long-term SA than the rest of the groups. The findings highlight the importance of expanding discourses on SA inequality to include aspects of occupation that are rarely discussed, such as occupational prestige. Interventions that would help employees in low prestige occupations to valorise their occupations may be helpful in promoting health and well-being and reducing SA, especially among men.

**Acknowledgements** We thank Ylva Ulfsdotter Eriksson for her constructive comments on this study.

**Contributors** CAN designed the study and was chiefly responsible for analysing the data and writing the manuscript drafts. GH participated in developing the research design, interpreting the results and writing the manuscript. TB delivered the data and contributed to the interpretation of the results and writing of the manuscript. All authors read and approved the final manuscript.

**Funding** This work was supported by (Forte) grant number (2016-07204).

**Competing interests** None declared.

**Patient consent for publication** Not required.

**Ethics approval** The regional ethical review board in Gothenburg approved the study (Dnr: 090-17). All the data used in this study were anonymised and the researchers do not have access to any personal information that could identify individuals included in the study.

**Provenance and peer review** Not commissioned; externally peer-reviewed.

**Data availability statement** Data may be obtained from a third party and are not publicly available. The authors do not have the permission to share the data that support the findings of this study due to data protection regulations. Deidentified participant data are available from Statistics Sweden (scb@scb.se) on reasonable request.

of the translations (including but not limited to local regulations, clinical guidelines, terminology, drug names and drug dosages), and is not responsible for any error and/or omissions arising from translation and adaptation or otherwise.

**ORCID iD**
Chioma Adanma Nwaru http://orcid.org/0000-0002-1772-2347

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
