## [Reviewer comments · BMJ Open]

ARTICLE DETAILS

TITLE (PROVISIONAL)	Occupational prestige and sickness absence inequality in employed women and men in Sweden: a registry-based study
AUTHORS	Nwaru, Chioma; Berglund, Tomas; Hensing, Gunnel

VERSION 1 – REVIEW

REVIEWER	Rahkonen, Ossi University of Helsinki, Department of Public Health
REVIEW RETURNED	02-Mar-2021

GENERAL COMMENTS	Socioeconomic (SEP) differences in sickness absence have been studied a lot. Several different measures on SEP have been used (occupational class, education, income, wealth, subjective social class etc). This paper is novel and innovative. It examines the association of occupational prestige and sickness absence and especially the gender-occupational prestige is emphasised. Often has been found that occupational differences in health among men are larger than among women. This paper looks this more carefully. It combines research on occupational differences and gender differences in health using intersectional approach. The paper is well done, fluently written and mainly clear analyses and results although we're dealing with an interesting but a somewhat complicated study and approach. Thus, I have quite a few comments that may help the authors when revising the paper. Most of the comments are minor. - My main question is, do we need those analyses where men and women are pooled?- My larger concern relates to the concept of prestige in occupations or occupational prestige. The authors emphasise only the psychological or psychosocial dimension and that working in the low prestige occupation affects your health via e.g. low self-esteem, alienation or boredom. It may, however, be (and really is) that the baseline for poor health is at least partly the physical work load of the occupations that have low prestige. Low prestige occupation often are physically hard occupations. Physically heavy work affects health directly and not via poor response to stress etc. Also low salary often means low prestige occupation. Income also affects and at least associates with health. I wouldn't mind seeing some discussion on this side of the coin.- If there is space the abstract may need as a first sentence the motivation of ratio of the paper. Each reader does not necessarily know what 'prestige' means in studies on occupation.- Related to the previous comment I'm not sure that all the readers understand similar way what 'occupational prestige' means. Perhaps the concept or at least the variable (the measurement) could be explained already in the abstract.
---

	- Introduction is well thought and written. The methods as well. The prestige groups could have a couple of examples of occupations for both women and men. Especially interested are the occupations in the high prestige occupations of women! Perhaps I read poorly but I didn't find any comment on 'previous SA', was it any SA or 15+ SA days and when ever or during last x years? Table 1 (design), I may have included marital status before socioeconomic measures. Who voted for the prestige for the occupations? The same respondents or some other population? Did the prestige order vary over time or did the occupations remain in the same groups over the study years? - The adjustment machine in model 1 is heavy, several and different factors are simultaneously adjusted for. The reader may like to see what kind of an effect age has on the association etc. I wouldn't miss crude figures, age-adjusted figures could be the first one. Previous SA may have very strong effect on the association? BTW, is that overadjustment because when you have heavy work you get ill and you receive SA certificate. Why to hide the previous work disability (SA) and work load when the occupations were not and are not similar in terms of effects on subsequent work ability. - page 17; It's somewhat unfair to state that 'Adjustment for employment-related variables did not seem to have much influence on the association', as there were not much to contribute after several sociodemographic and socioeconomic variables in addition to previous SA were already adjusted for. Did the authors think to first adjust for the variables separately? It's not possible to show all those analyses in the table, but it's good to know what effects they have. - After seeing the results of Table 3, I don't see any reason why the gender-pooled analyses are presneted. Could these analyses (Table 2 & table 4) be omitted from the paper? Results from the table 3 really are surprising. I'm sure the authors have checked those results many times. If Table 2 and table 4 are omitted I would prefer seeing a new and larger Table 1 by gender. - Strengths and weaknesses are properly discussed. Perhaps a comment on physical work load may be included. More minor comments: - gender and sex are mixed, at least once I found 'sex' (page 24). Perhaps for reason - References, 37, should be LaaKsonen M,
--	--

REVIEWER	Kang, Mo-Yeol College of Medicine, The Catholic University of Korea, Department of Occupational and Environmental Medicine
REVIEW RETURNED	01-Apr-2021

GENERAL COMMENTS	This study investigates the association between occupational prestige and sickness absence, and the distribution of the association in women and men, and the effect of intersections of gender and occupational prestige on sickness absence. The methodology, results and interpretations seem sound. The paper is well written. There are some major limitations of the study, but these are well described in the discussion section. So, I found merit of this study and I think this article is acceptable with minor revision. Below, please find my comments and suggestions. I hope you find them contributing to your research. 1) I found some parts of introduction are unnecessary. Please tighten the introduction to focus the rationale for the study question
--

	within the context of the role of occupational prestige as a determinant of SA. 2) Page 6, line 1-13: Authors might have missed “)” in the sentence below: ‘Investigating this association can help to capture features of social hierarchy (acceptance, respect, recognition, admiration, autonomy, power, social network and social support [10, 11] that may not be adequately represented in other measures of occupational classification, such as occupational class, thus illuminating how social position and its associated rewards and privileges contribute in shaping health risk.’ 3) Page 8: I think a figure would be helpful to understand how final participants of this study was selected. 4) Page 11: I wonder if analysis using GEE with logistic regression assume that long-term SA had happened only once in a year or not. Please clarify it. 5) Page 11: Long-term SA was defined as absence days lasting over 120 days, but authors conducted GEE with logistic regression using 90 days as the cut point. It is somewhat confusing to the reader outside of Sweden. Please explain it in more detail. 6) Table 1: In column head, “Incidence rate ratios (IRR) associated with number of SA† days at follow-up”, I wonder if “†” is also valid here.
--	---

VERSION 1 – AUTHOR RESPONSE

Reviewer: 1

Dr. Ossi Rahkonen, University of Helsinki

Comments to the Author:

Socioeconomic (SEP) differences in sickness absence have been studied a lot. Several different measures on SEP have been used (occupational class, education, income, wealth, subjective social class etc). This paper is novel and innovative. It examines the association of occupational prestige and sickness absence and especially the gender-occupational prestige is emphasized. Often has been found that occupational differences in health among men are larger than among women. This paper looks this more carefully. It combines research on occupational differences and gender differences in health using intersectional approach. The paper is well done, fluently written and mainly clear analyses and results although we’re dealing with an interesting but a somewhat complicated study and approach. Thus, I have quite a few comments that may help the authors when revising the paper. Most of the comments are minor.

Response: We thank the reviewer for the appreciation of our study and its importance. We are grateful for thorough and valuable comments.

Comment 1: My main question is, do we need those analyses where men and women are pooled?

Response: Since this is the first study on this topic, we think that it is important to present the pooled analyses with the overall association. Thus, we believe that the pooled analyses provide valuable information to understanding the association in general.

Comment 2: My larger concern relates to the concept of prestige in occupations or occupational prestige. The authors emphasize only the psychological or psychosocial dimension and that working in the low prestige occupation affects your health via e.g. low self-esteem, alienation or boredom. It may, however, be (and really is) that the baseline for poor health is at least partly the physical work load of the occupations that have low prestige. Low prestige occupation often are physically hard occupations. Physically heavy work affects health directly and not via poor response to stress etc. Also low salary often means low prestige occupation. Income also affects and at least associates with health. I wouldn't mind seeing some discussion on this side of the coin.

Response: We agree with the reviewer's comment that lower prestige occupations often are physical heavy work and that that might have a direct effect on their health. For lack of data, we could not evaluate the influence of physical workload on the association between occupational prestige and SA. We have now added a sentence in the discussion part (page 37) about an increased risk of high physical workload among employees in lower social positions and the consequent effect of that on SA. In addition, we have clearly stated, as a limitation, our inability to assess the influence of physical workload on the association (page 37).

Regarding the effect of income, we did say, in the discussion part (page 34), that income, as well as education and occupational class, attenuated the risk estimates; however, they did not explain away the association between occupational prestige and sickness absence.

Comment 3: If there is space the abstract may need as a first sentence the motivation of ratio of the paper. Each reader does not necessarily know what 'prestige' means in studies on occupation.

Response: We have now included the rationale for the study in the abstract. We have also added a definition of occupational prestige (page 2).

Comment 4: Related to the previous comment I'm not sure that all the readers understand similar way what 'occupational prestige' means. Perhaps the concept or at least the variable (the measurement) could be explained already in the abstract.

Response: We have added a definition of occupational prestige in the abstract (page 2).

Comment 5: Introduction is well thought and written. The methods as well. The prestige groups could have a couple of examples of occupations for both women and men. Especially interested are the occupations in the high prestige occupations of women! Perhaps I read poorly but I didn't find any

comment on 'previous SA', was it any SA or 15+ SA days and whenever or during last x years? Table 1 (design), I may have included marital status before socioeconomic measures. Who voted for the prestige for the occupations? The same respondents or some other population? Did the prestige order vary over time or did the occupations remain in the same groups over the study years?

Response: We have now provided examples of high prestige occupations (page 12) in women and men in our study. Regarding previous SA, we did state in the methods section (page 10) that previous SA referred to SA 2 years preceding the study observation. We have now added this information to the table to enhance clarity. We have also moved information on marital status before socioeconomic measures (table 2) as suggested (pages 17-19).

We measured occupational prestige using Treiman's SIOPS, which was calculated based on occupational prestige ratings from over 60 countries (Treiman, 1976). Although Sweden did not participate in the study, a Swedish study that compared the SIOPS and scores generated from a Swedish population found a high correlation (0.8) between the scores. We have now added this information in the method's section of our paper (page 9).

It may be that some participants have changed occupations and consequently moved places in the occupational prestige scale. Unfortunately, we did not have data to assess whether an individual's occupational prestige status remained or changed during the study follow-up. We, therefore, have included this information as one of the study limitations (page 37).

Comment 6: The adjustment machine in model 1 is heavy, several and different factors are simultaneously adjusted for. The reader may like to see what kind of an effect age has on the association etc. I wouldn't miss crude figures, age-adjusted figures could be the first one. Previous SA may have very strong effect on the association. BTW, is that overadjustment because when you have heavy work you get ill and you receive SA certificate. Why to hide the previous work disability (SA) and work load when the occupations were not and are not similar in terms of effects on subsequent work ability.

Response: We have revised the analyses and now provided age-adjusted figures, as well as a separate model (Model II in the tables) to clearly show the effect of previous sickness absence.

Comment 7: page 17; It's somewhat unfair to state that 'Adjustment for employment-related variables did not seem to have much influence on the association', as there were not much to contribute after several sociodemographic and socioeconomic variables in addition to previous SA were already adjusted for. Did the authors think to first adjust for the variables separately? It's not possible to show all those analyses in the table, but it's good to know what effects they have.

Response: We agree with the reviewer's comment that the statement was a bit unfair. We have now modified the sentence and state that "Adjustment for employment-related variables did not seem to have much influence on the association after adjustment for sociodemographic and socioeconomic

factors”(pages 21,22). We hope that this modification would help to clarify why employment-related variables may not have had any effects on the association.

Indeed, it would be interesting to show the effects of each study variable on the association. However, because the current analyses do not primarily focus on the mechanism behind the association, we would want to restrict our focus on the study aim, which is to establish if occupational prestige has an independent influence on sickness absence. In a future ongoing study, we have added a new register with more detailed information on employment-related variables, including psychosocial work environment factors. In that study, we hope to delve into understanding the mechanism underlying the association observed in this study. We believe that that study will give clarity on the effects of different factors more closely related to the work and employment situation.

Comment 8: After seeing the results of Table 3, I don't see any reason why the gender-pooled analyses are presented. Could these analyses (Table 2 & table 4) be omitted from the paper? Results from the table 3 really are surprising. I'm sure the authors have checked those results many times. If Table 2 and table 4 are omitted I would prefer seeing a new and larger Table 1 by gender.

Response: We would want to keep tables 2 and 4 in the study because we think that they contribute valuable knowledge to the topic in general. We have made some changes: presented table 2 and provided table 4 as text. In addition, we think that providing gender-stratified table 1 would overshadow the core objective of the paper that is to establish whether occupational prestige was associated with sickness absence. For this reason, we would want to retain table 1 (now table 2) in its current state.

Comment 9: Strengths and weaknesses are properly discussed. Perhaps a comment on physical workload may be included.

Response: We have now included, as a limitation of the study, the lack of data on physical workload (page 37)

More minor comments:

Comment 10: Gender and sex are mixed, at least once I found 'sex' (page 24). Perhaps for reason

Response: Corrected

Comment 11: References, 37, should be LaakSonen M,

Response: Corrected. We thank the reviewer for pointing out this oversight.

Reviewer: 2

Prof. Mo-Yeol Kang, College of Medicine, The Catholic University of Korea

Comments to the Author:

This study investigates the association between occupational prestige and sickness absence, and the distribution of the association in women and men, and the effect of intersections of gender and occupational prestige on sickness absence. The methodology, results and interpretations seem sound. The paper is well written. There are some major limitations of the study, but these are well described in the discussion section. So, I found merit of this study and I think this article is acceptable with minor revision. Below, please find my comments and suggestions. I hope you find them contributing to your research.

Response: We thank the reviewer for the positive comment and for a thorough and valuable review.

Comment 1: I found some parts of introduction are unnecessary. Please tighten the introduction to focus the rationale for the study question within the context of the role of occupational prestige as a determinant of SA.

Response: We have now revised the introduction (pages 5-7).

Comment 2: Page 6, line 1-13: Authors might have missed “)” in the sentence below:

‘Investigating this association can help to capture features of social hierarchy (acceptance, respect, recognition, admiration, autonomy, power, social network and social support [10, 11] that may not be adequately represented in other measures of occupational classification, such as occupational class, thus illuminating how social position and its associated rewards and privileges contribute in shaping health risk.’

Response: Corrected. We thank the reviewer for pointing out this oversight.

Comment 3: Page 8: I think a figure would be helpful to understand how final participants of this study was selected.

Response: We have now provided a flowchart illustrating the selection of study participants.

Comment 4: Page 11: I wonder if analysis using GEE with logistic regression assume that long-term SA had happened only once in a year or not. Please clarify it.

Response: It is possible for an individual to have more than one episode of long-term (>120 days) sickness absence during the three-year follow-up, and GEE with logistic regression captures the repeated episodes. We have now clarified this in the text (page 15)

Comment 5: Page 11: Long-term SA was defined as absence days lasting over 120 days, but authors conducted GEE with logistic regression using 90 days as the cut point. It is somewhat confusing to the reader outside of Sweden. Please explain it in more detail.

Response: We have modified the sentence in the statistical section to state that “The robustness of the results regarding long-term SA was tested by repeating the analysis using 90 days as the cut point for long-term SA since the Swedish evaluation of work capacity within the rehabilitation framework starts at day 90” (page 15). We hope that this modification provides clarity to the text.

Comment 6: Table 1: In column head, “Incidence rate ratios (IRR) associated with number of SA† days at follow-up”, I wonder if “†” is also valid here.

Response: Corrected

Reviewer: 1

Competing interests of Reviewer: None declared

Reviewer: 2

Competing interests of Reviewer: I declare no competing interest regarding this paper.

VERSION 2 – REVIEW

REVIEWER	Kang, Mo-Yeol College of Medicine, The Catholic University of Korea, Department of Occupational and Environmental Medicine
REVIEW RETURNED	21-May-2021
GENERAL COMMENTS	The authors have addressed all comments and made appropriate revisions and corrections to the revised manuscript. The revised one is much improved so that I have no more comments to be revised for the article.